# Compact and Efficient Topological Mapping for Large-Scale Environment with Pruned Voronoi Diagram

**Yao Qi [1], Rendong Wang [2,*], Binbing He [2], Feng Lu [1,2] and Youchun Xu [2]**

[1] Department of Vehicle Engineering, Army Military Transportation University, No.1 Chenglin Street, Tianjin 300161, China; qii9471@163.com (Y.Q.); lf_0623@163.com (F.L.)

[2] Institute of Military Transportation, Army Military Transportation University, Tianjin 300161, China; jjuv_hebinbing@163.com (B.H.); xu56419@126.com (Y.X.)

\* Correspondence: wrd1992@163.com

**Abstract:** Topological maps generated in complex and irregular unknown environments are meaningful for autonomous robots' navigation. To obtain the skeleton of the environment without obstacle polygon extraction and clustering, we propose a method to obtain high-quality topological maps using only pure Voronoi diagrams in three steps. Supported by Voronoi vertex's property of the largest empty circle, the method updates the global topological map incrementally in both dynamic and static environments online. The incremental method can be adapted to any fundamental Voronoi diagram generator. We maintain the entire space by two graphs, the pruned Voronoi graph for incremental updates and the reduced approximated generalized Voronoi graph for routing planning requests. We present an extensive benchmark and real-world experiment, and our method completes the environment representation in both indoor and outdoor areas. The proposed method generates a compact topological map in both small- and large-scale scenarios, which is defined as the total length and vertices of topological maps. Additionally, our method has been shortened by several orders of magnitude in terms of the total length and consumes less than 30% of the average time cost compared to state-of-the-art methods.

**Keywords:** topological maps; generalized Voronoi diagram; largest empty circle; unstructured environment; large-scale environment



## 1. Introduction

As an important achievement of SLAM (Simultaneous Localization and Mapping), topological maps play an important role in the environment representation [1,2]. In contrast from a metric map and a semantic map, a topological map uses a skeleton composed of vertices and edges to describe the world. The skeleton makes the topological map compact and efficient, which is illustrated by the lower memory consumption of storage and lower time cost of map updating and routing, especially in large-scale scenarios [3]. These advantages of topological mapping are vital in geographic environment modeling, autonomous exploration, and robot navigation [4].

The topological representation problem is similar to the robot path-planning problem [5]. It focuses on the multi-path, while path planning aims to find the optimal result. The three most established methods are random sampling-based, visibility graph-based and skeleton-based, which can solve the problem in known, partially known, and unknown environments.

Random sampling-based method. The classic methods using random sampling include the probabilistic roadmap (PRM) [6], sparse roadmap spanners (SPARS) [7], and rapidly exploring random tree (RRT) family [8], which include RRT and its variants, such as Bi-RRT, RRT*, and BIT* [9,10]. Witting constructed the topological map using a history of previously visited positions as seeds of the RRT to guide MAVs' autonomous exploration [11]. Additionally, Umari used multiple RRTs [12]. TARE constructed a sparse

random roadmap in the traversable space expanded from the past trajectory for global planning [13]. To adapt to a large environment, Wang and Dang incrementally constructed a graph structure along with the exploration process by sampling random points uniformly [14,15]. These methods are complete when they sample enough points, which requires high computation in narrow environments.

Visibility graph-based method. The method constructs visibility edges among obstacles' vertices, which need to extract convex polygons from irregular obstacles [16], and sometimes the extraction is complex in unstructured environments. The FAR expanded the visibility edges incrementally in dynamic scenarios and addressed navigation tasks in both known and unknown environments [17]. Lee and Shah used the Visibility Graph (VG) for long-distance planning for USVs fulfilling target missions in marine environments [18,19]. Most of them use A* to efficiently search for the optimal path in VG. The limits of VG in complex environments are its high computational complexity and the requirement for well-defined polygonal geometry.

Skeleton-based method. Thrun first proposed generalized Voronoi diagram (GVD)-based topological maps and segmentation [20], which can be used for the safe autonomous navigation of mobile robots in unknown structured environments [21]. Li constructed a reduced approximated generalized Voronoi graph (RAGVG) by an occupancy grid map (OGM), which improved autonomous exploration [22]. Ok considered uncertainty with GVD [23]. The Voronoi edge is the perpendicular bisector of adjacent sites, and thus, the map paths are safer than other topological map construction methods [24]. The methods to construct GVD incrementally include dynamic brushfire [25] and a random incremental algorithm [26]. Most dynamic brushfire algorithms must load the whole range area first and calculate the Euclidean distance maps (EDM), just as the Dijkstra searcher. Liu constructed Voronoi segmentation incrementally with virtual obstacles using dynamic brushfire [3]. The random incremental algorithm is an efficient Voronoi diagram generation method in computational geometry by adding sites one by one, but it is poor at removing the existing site [27].

Constructing a skeleton to describe the reachable region is the main challenge of topological mapping. Some methods are efficient enough, but the maps generated by them are not compact enough. For example, visibility graph-based methods easily generate too many road branches, while RRT- or PRM-based methods easily generate redundant vertices and edges on the trunk road. In this article, our topological map is built based on a Voronoi diagram, which is generated by obstacle information in a metric map. Considering the increase in time and decrease in accuracy caused by obstacle clustering and polygon extraction, our Voronoi diagram is generated based on the original obstacle points directly. On this basis, a new pruning strategy is proposed that deletes unnecessary branches while retaining the main road, and the strategy is used to simplify the relationship in the main road as much as possible.

Many existing methods of topological mapping based on Voronoi diagrams are one-time or global incremental. In other words, these methods have to update the whole map when the environment changes slightly, rather than updating the changed region. The efficiency of these methods decreases substantially with the expansion of the environmental scale, and it is difficult to build and update the topological map in real-time. To solve this problem, we propose a local incremental updating method as shown in Figure 1a, which can update topological boundaries only within a changed region. The method is implemented by using the largest empty circle property of the Voronoi diagram. Thus, the real-time and high efficiency of map construction are ensured.

The benefit of our method is that it is both more compact and more efficient than the state-of-the-art methods. We evaluate the proposed incremental and pruned Voronoi mapper (IPVM) by comparing it against state-of-the-art and classical methods in small- and large-scale simulation benchmarks. The results conclude that the proposed method generates fewer vertices and shorter map lengths. Furthermore, we also validate the online

topological map generator in a real-world experiment under some posed drifts. The main contributions of this paper are summarized as follows:

Compactness: We propose a novel Voronoi diagram pruning strategy, which can extract the road skeleton in the environment in a sufficiently compact way.

Efficiency: Based on the largest empty circle property of the Voronoi diagram, we propose a novel method to incrementally update the topological map in the changed region rather than the whole map. Even in large-scale and complex environments, the map can still maintain efficiency in real-time.

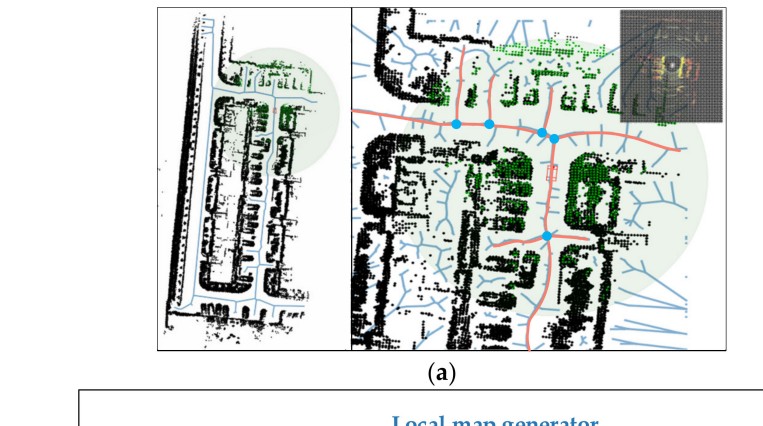

**(a)**

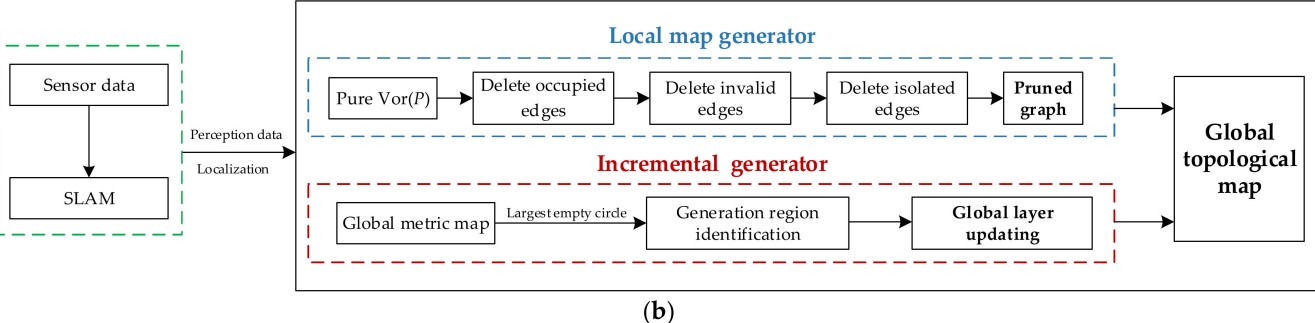

**(b)**

**Figure 1.** An overview of framework. (**a**) Global and local topological structure. The black areas are global occupied grids, and the nattier blue lines are edges of the graph; the pink edges are extracted by our algorithm, and the blue circles are vertices of the routing graph. The point cloud shown in the upper right is new data corresponding to occupied grids. A local map is generated from the metric map using a pruned Voronoi diagram and is merged into the global structure. (**b**) Flow diagram of the proposed topological mapping framework.

As illustrated in Figure 1b, the proposed work operates upon perception data and localization data, and it is composed of a local topological map generator (Section 2) and an incremental map generator (Section 3). This paper is structured as follows: In Sections 2 and 3, we introduce our proposed method in detail. Experimental results and analyses are presented in Section 4. Our work is summarized in Section 5.

## 2. Pruned Voronoi Graph Construction

### 2.1. Voronoi Diagram

The traditional Voronoi diagram is the spatial division of point sets, also known as the Thiessen polygon. We define points set (sites) $P = \{P_1, P_2, \cdots, P_N\}$, $d(P_i, P_j)$ as the Euclidean distance between point $P_i$ and $P_j$, $C(P_i)$ is the matched Voronoi cell of site $P_i$, and $\text{Vor}(P)$ denotes the Voronoi diagram of sites $P$. The $C(P_i)$ is defined as follows:

$$C(P_i) = \{x | d(x, P_i) < d(x, P_k),\ P_i, P_k \in P, i \neq k,\ i, k \in \{1, 2, \cdots, N\}\} \tag{1}$$

### 2.2. Pruned Voronoi Graph

The purpose of establishing a Voronoi diagram in this paper is to obtain the topological structure of the environment. The skeleton structure of the topological map is GVD, which divides space into generalized Voronoi cells around objects $O$ [28]. However, irregular objects are difficult to define $d(O_i, O_j)$. Based on the traditional Voronoi construction method [24] and our pruning method, an approximate GVD is generated. We define $E_D = \{E_1, E_2, \ldots, E_M\}$ and $V_D = \{V_1, V_2, \ldots, V_N\}$ as the edges and vertices of Vor($P$), where $E_{i,j} = (V_i, V_j)$, $i, j \in \{1, 2, \cdots, N\}$, and $L(V_i, V_j)$ denotes the line segment consisting of two vertices. We dilate the environment to configuration space $C = C_{\text{free}} \cup C_{\text{occupied}}$ and then eliminate the invalid $E_{i,j}$ if $L(V_i, V_j) \cap C_{\text{occupied}} \neq \varnothing$, as shown in Figure 2.

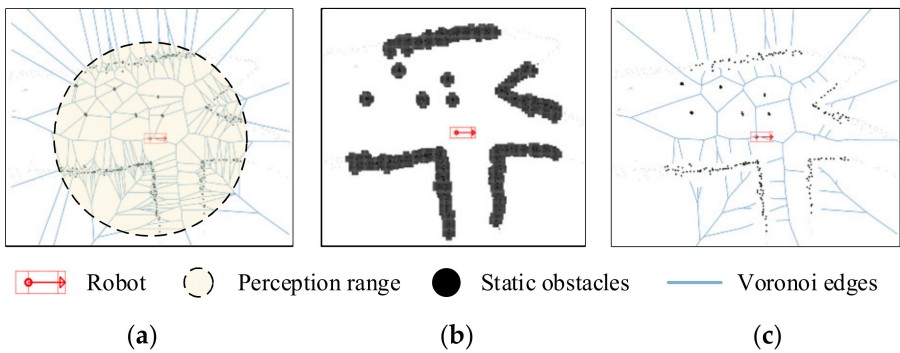

| | | |
|---|---|---|
| Robot | Perception range | Static obstacles | Voronoi edges |

(**a**)  (**b**)  (**c**)

**Figure 2.** Pruned Voronoi diagram. (**a**) Origin Voronoi diagram generated by [24], (**b**) $C$ space with an obstacle dilated, and (**c**) Voronoi diagram pruned by $C_{\text{occupied}}$.

Then, the undirected graph $G = (V_G, E_G)$ is constructed by the geometrical relationship of adjacent edges in Vor($P$), and $V_{Gj}$ and $V_{Gp}$ are connected if

$$L(V_{Gi}, V_{Gj}) \cap L(V_{Gp}, V_{Gq}) \neq \varnothing (i \neq j, p \neq q) \tag{2}$$

The undirected Voronoi graph has many invalid edges that are short and isolated. We reduce those edges by two steps and obtain a reduced approximated generalized Voronoi graph (RAGVG) for further pruning. First, we find the edge $E_{i,j}$ that has only one joint relation in the set $E_G$ except itself, in other words, the leaf node. We eliminate $E_{i,j}$ by the robots' length $d_v$ and distance buffer $d_{\text{buffer}}$. Therefore, we regard the short edge $E_{i,j}$ as an invalid edge by Equation (3) as follows

$$\begin{cases} E_{i,j} \cap E_G = \left\{ E_{p,q} \middle| (p = i \wedge q \neq j) \vee (p = j \wedge q \neq i) \right\} \\ d(i,j) \leq d_v + d_{\text{buffer}} \end{cases} \tag{3}$$

Second, this paper divides the topological structure $G$ into two segmentations by topological relation: The main segmentation $S_m$ where the robot is located and the isolated segmentation $S_s$, and $S_s \cap S_m = \varnothing$. In the divided algorithm, we select the closest edge $E_r$ with the robot location in the history graph and then add $E_r$ into the open list L and the priority queue Q. Aiming at traversing the whole $G$ quickly, we ignore the distance of each edge and apply the uninformed search strategy known as the breadth-first search (BFS) algorithm until Q is empty. The vertices that have searched in BFS consist of the main segmentation $S_m$, and others consist of $S_s$. We eliminate the $S_s$ that cannot arrive from the robot location in $G$, as shown in Figure 3. We also simplify the adjacent edges with similar angles to an edge for fast route planning.

After the above steps, we obtain the RAGVG from the undirected Voronoi graph, and the final $G$ is a connected graph, enabling the route path to be found at any starting point and ending point within the map. Through these steps, we do not need to cluster and extract the polygon of obstacles in complex and irregular environments. Hence, the

algorithm reduces the error of geometric shape fitting and the complexity of constructing the GVD.

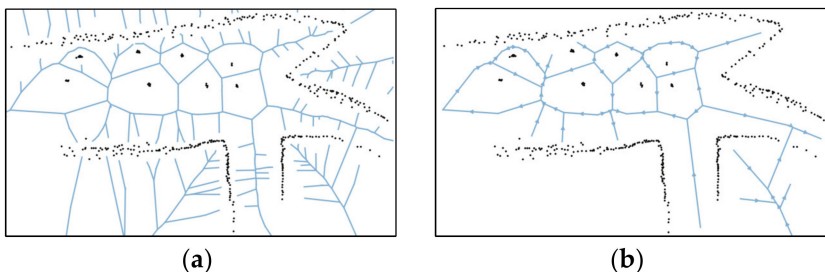

**Figure 3.** RAGVG construction. (**a**) The undirected Voronoi graph, and (**b**) the RAGVG constructed by BFS and simplification.

## 3. Incremental Topological Map Generator

Due to the continuous growth of the map region, the computation of RAGVG for the whole region will consume a large amount of resources. Based on the results in Section 2, we construct the local layer graph $G_{local}$, and then we merge the $G_{local}$ into the global layer graph $G_{global}$ to update the overlapping area in the whole map. The Voronoi vertices and edges are affected by added or removed sites and considering that obstacle points do not increase or decrease randomly in the global map but change in a fixed local area close to the robot's location, we replace the $G_{local}$ in region $R_N$ by the vertex property of the largest empty circle.

### 3.1. Perception Data Update

We divide sites in blocks by geometrical position and place them into file storage in a large scenario. The perception result of the latest frame input is overlapped with the historical map based on the grid index. Obstacle points in the same grid have the same index, as shown in Figure 4. There are also some useless points (grid points of dynamic obstacles), which will be deleted according to the attribute identification from the perception results.

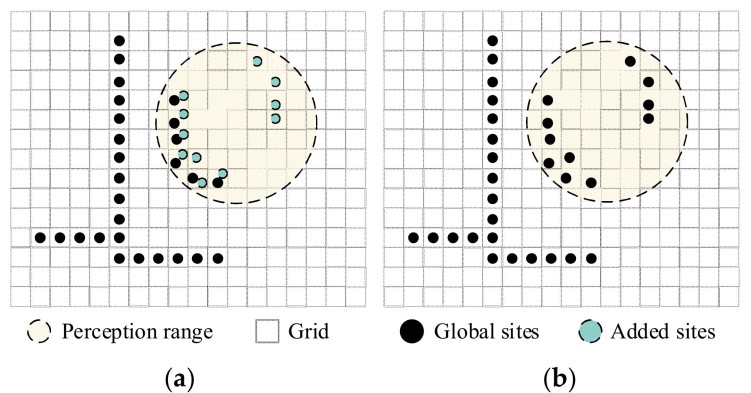

**Figure 4.** Global metric map update. (**a**) The global map and existing sites. (**b**) The updated map with the latest data.

### 3.2. Voronoi Diagram Generation Region Identification

With $q$ as the center of a circle, the largest circle without any sites in $P$ is called the largest empty circle $C_P(q)$.

**Theorem 1.** *$V_q$ is a vertex in Vor(P) if and only if there are at least three sites on the boundary of its largest empty circle $C_P(V_q)$, as illustrated in Figure 5a.*

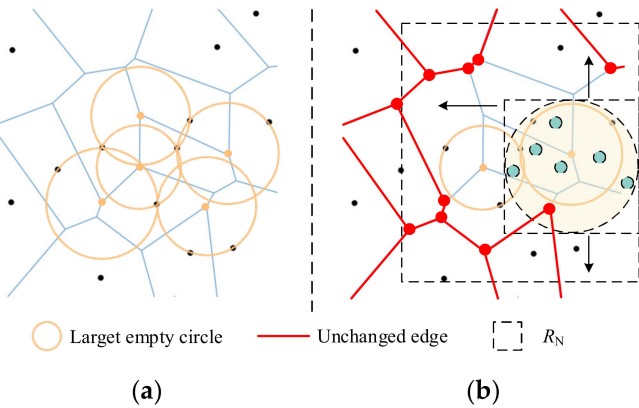

Large empty circle ——— Unchanged edge   $R_N$

(a)                                    (b)

**Figure 5.** Voronoi diagram generation region. (**a**) The largest empty circle of Voronoi vertices. (**b**) A schematic diagram of a region identified by Theorem 1.

The change in the Voronoi diagram is the vertices' position change essentially, and vertices' positions are influenced by adjacent sites. The sweep line algorithm generates Voronoi edges by constantly moving a fixed line [24]. Inspired by this idea, we design a minimum area identification algorithm based on the sweep line to obtain $R_N$. The edges inside of the minimum polygon $R_N$ are changed or removed after the latest points $P_N$ update, while the edges and vertices outside of $R_N$ are consistent. We obtain the whole sites $P = P_N \cup P_H$ within the perception range first, where $P_H$ is history sites and $P_N$ denotes new sites. Then, we obtain the rectangular envelope box $R_{min}$ *of P* and sweep the side-line. During line sweeping, edges $E_L$ that satisfy $E_L = E \cap L_i$ are found, and vertices $V_L = \{V_1, V_2, \ldots, V_N\}$ of $E_L$ are contained in $R_N$. The sweeping stops when $\forall p$ in $N$, and the following exists

$$\begin{cases} P_C = C_P(V_q) \cap P \\ P_C = \{s_k | k \in \{1, 2, \cdots, K\}\} \\ K \geq 3 \land \forall k, s_k \in P_H \end{cases} \qquad (4)$$

where $P_C$ is the sites within or on the circumference of $C_P(V_q)$. Note that vertex $V_q$ does not change when any site $s_k$ belongs to historical site $P_H$. In practice, we use the k-d tree to find the nearest site of $V_p$ and its distance $d_k$, which is the radius of $C_P(V_q)$. Then, we search sites $s_k$, of which the distance to $V_q$ are shorter than $d_k$ in the k-d tree. According to Theorem 1, if the size of $P_C$ is greater than 2 and all the $s_k$ belong to $P_H$, $V_p$ will not be influenced by $P_N$.

*3.3. Global Layer Update*

We divide the whole space into two subspaces through the above steps, and the subspace $R_N$ aims to store the surfaces of potential change after the perception data update. Again, we extend the rectangle to contain the sites belonging to $C_P(V_q)$ and set the distance buffer $d_B$ to obtain the final region $R_{max}$, as seen in Figure 6. The Voronoi diagram $Vor(P_N)$ is generated and pruned by sites within $R_{max}$, and then we remove edges out of $R_N$. It should be noted that the edge is out of $R_N$ if and only if two vertices are not within $R_N$. In other words, we remove vertices out of $R_N$ in $Vor(P_N)$. Finally, we remove the obsolete vertices in $Vor(P)$ within $R_N$ and fulfill the subspace with $Vor(P_N)$ to update the global layer map.

In practice, we hold two graph structures in the global layer: $Vor(P)$, which is pruned by Section 2 and constructed incrementally by Section 3, and $G_{global}$, which is constructed incrementally from $Vor(P)$. Let $n \in \mathbb{Z}^+$ be the number of points in $R_N$. The time complexity of constructing $Vor(P_N)$ is $O(n\log n)$, and $R_N$ identification consumes $O(n)$ time. Benefitting from storing data in geometrical blocks, concatenating $Vor(P)$ and $Vor(P_N)$ takes constant time. The time complexity of the global layer map update algorithm is $O(n\log n)$. Similar to

other route-planning algorithms, we use A* to search in the topological map $G_{global}$ for the shortest route path.

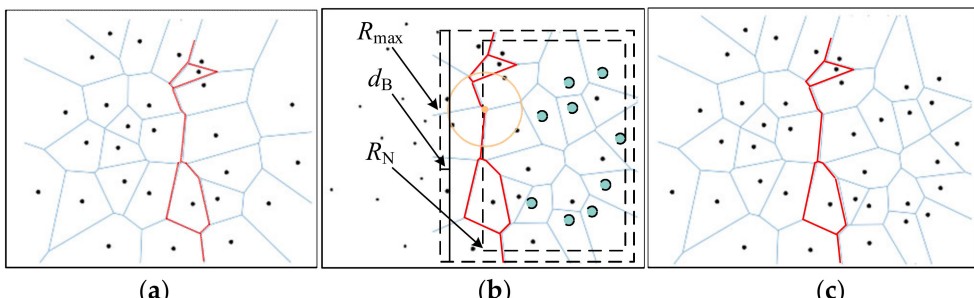

**Figure 6.** Global Voronoi diagram updated incrementally. (**a**) The origin Vor($P_H$). In (**b**), we identify $R_N$ and $R_{max}$ in the origin Vor($P_H$) and generate Vor($P_N$). (**c**) The updated Voronoi diagram.

## 4. Results

We conduct experiments in both benchmark simulation and the real world. The benchmark is from the autonomous exploration development environment [29]. Our data collection platform in the real world is equipped with a 32-line LiDAR forward and a 16-line LiDAR backward. In the simulation, we do not consider the large location error and the closed-loop correction in SLAM mapping. The mapper algorithm runs on a 3.2 GHz, i7-8700 computer using a single CPU thread, and all methods replan at 1 Hz. We compare our method against four representative methods, all using open-source code to experiment on the topological mapper: TARE [13]: A method using random sampling to obtain topological maps for fast autonomous exploration; FAR [17]: A visibility graph-based method for fast route planning in known and unknown environments. These two methods generate maps incrementally. Additionally, we test an incremental Voronoi Diagram generator [25], which employs a dynamic variant of the brushfire algorithm to update those cells that are actually affected by the environmental change (abbreviated as DB) and a pure Voronoi diagram generator: Fortune's sweep line algorithm with the whole obstacle points [24].

### 4.1. Benchmark Experimental Results

We conduct simulation experiments in indoor (130 m × 100 m), tunnel (330 m × 250 m), and forest (150 m × 150 m) areas [29]. These scenarios include indoor and outdoor, sparse and dense, and different ranges and topological structures. We input the map size in DB and prune all Voronoi diagrams by our algorithm in Section 2. Then, we only record the time costs of DB and Fortune. The perception range is approximately 20 m, and the grid size is 0.2 m × 0.2 m in the perception data update.

In the simulated experiments, we use TARE to explore the whole environment. Figure 7a shows topological maps of the three methods in the indoor scenario. DB and Fortune have almost the same maps as ours. To quantify the efficiency and ability of environment representation, we record the time cost in each frame in Figure 7b, the total length of all edges, and the total number of vertices as the explored areas grow in Figure 7c,d. The results indicate that we have the lowest time cost in most frames and the shortest length. Our results have flat surfaces and sometimes decrease as the travel distance grows in Figure 7c,d, because an increasing number of redundant vertices are pruned when the metric map tends to be completely enclosed.

We also compare methods quantitatively in larger and outdoor environments, as shown in Figure 8. Note that compared with the indoor environment, these two environments have many more branches and more irregular obstacles. All the benchmarked methods are able to generate maps that approximately cover the entire environment without a time limit. The maps constructed by TARE and FAR have many redundant edges and vertices, especially in the unstructured forest with visibility graph-based FAR, while TARE

has insufficient sampling in some corners for the time limitation, as shown in Figure 7a. Our method prunes the invalid edges and obtains the near-centerline of adjacent regions, and thus, we obtain more comprehensible and lightweight maps.

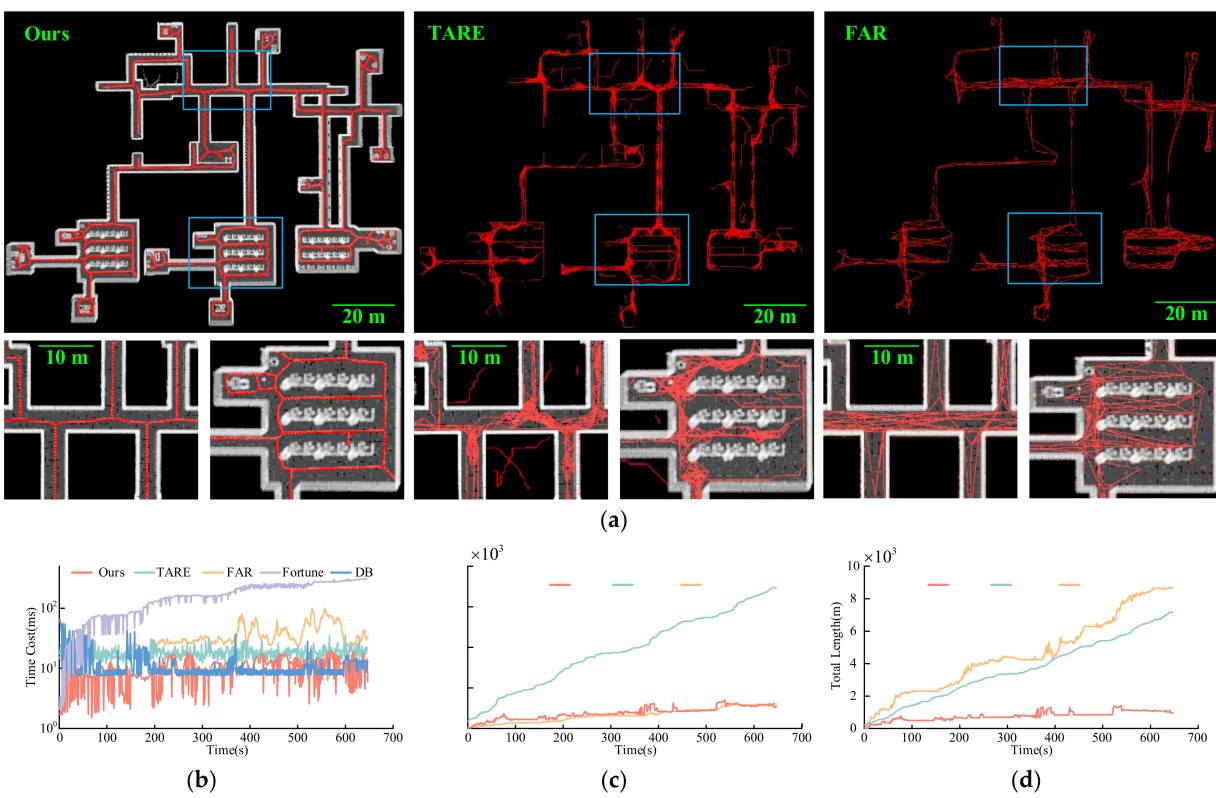

**Figure 7.** Benchmark comparison of the proposed method, TARE, FAR, Fortune, and DB in an indoor space. (**a**) The resulting map and a partially enlarged drawing of the two intersections. (**b**) The time cost of constructing the map in a single frame over the exploration time duration, and (**c**,**d**) the total number of vertices and the total length of all edges.

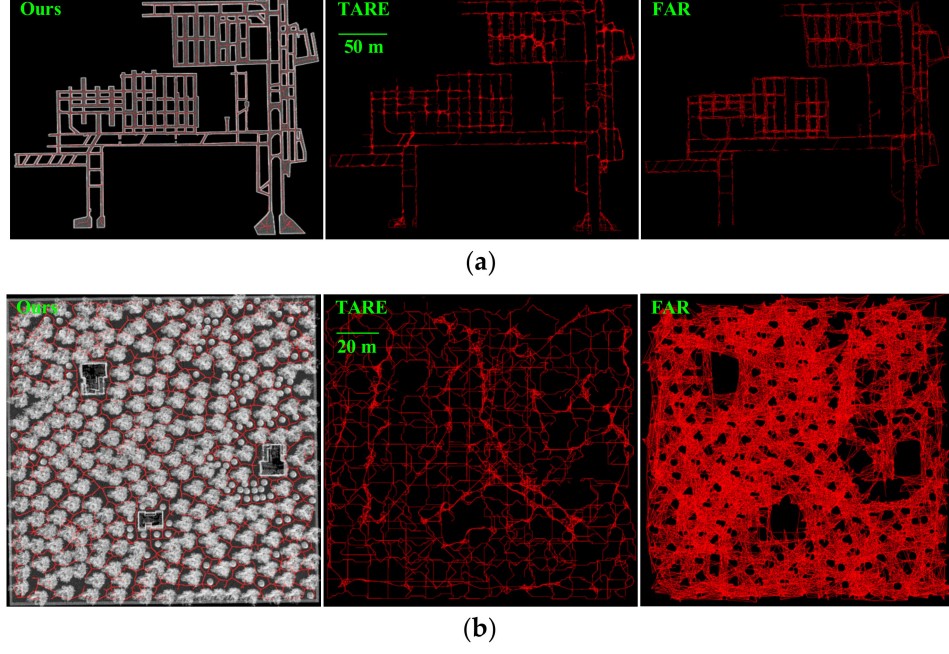

**Figure 8.** Resulting maps. (**a**) In the tunnel, and (**b**) in the forest.

Table 1 denotes the average time cost of the five methods. Fortune's algorithm is the most efficient in generating a pure Voronoi diagram, but without generating incrementally, it costs more and more time in global layer graph construction. DB must know the scene's size to establish a grid map and costs more time when facing a larger environment, such as a tunnel. Our method is slightly influenced by the number of obstacles in the environment, so the average time cost increases in the forest and tunnel. The average time cost of our method in all scenarios decreases by 30% compared to DB, 37% compared to TARE, and is much shorter than that of the other methods.

**Table 1.** Average time cost.

| Scenario | Fortune | DB | TARE | FAR | Ours |
|---|---|---|---|---|---|
| indoor (ms) | 169.48 | 11.73 | 18.16 | 32.51 | 9.86 |
| tunnel (ms) | 3826.27 | 26.56 | 25.13 | 43.87 | 13.20 |
| forest (ms) | 1214.45 | 13.59 | 15.30 | 706.06 | 13.70 |
| average (ms) | 1376.8 | 17.30 | 19.53 | 260.81 | 12.25 |

Tables 2 and 3 present the simulation results in map metrics. Although the forest area is smaller than the tunnel, many more obstacles lead to additional vertices. The proposed method establishes a more compact connection than the other methods with fewer vertices in Table 2. The other point we can see is that our maps' total length is over several orders of magnitude shorter than the other methods' in Table 3, especially in the complex forest. The result indicates the strength of our framework in generating compact topological maps.

**Table 2.** Total vertices of maps.

| Scenario | TARE | FAR | Ours |
|---|---|---|---|
| indoor | 1737 | 301 | 267 |
| tunnel | 5641 | 928 | 781 |
| forest | 3513 | 4427 | 2801 |

**Table 3.** Total length of maps.

| Scenario | TARE | FAR | Ours |
|---|---|---|---|
| indoor (km) | 7.12 | 8.65 | 0.96 |
| tunnel (km) | 26.28 | 50.45 | 5.23 |
| forest (km) | 13.83 | 213.871 | 4.83 |

*4.2. Physical Experimental Results*

To further validate the proposed method, we conduct field experiments in residential areas (100 m × 100 m), and the trees on both sides of the road give some posed drifts. The experimental environment and associated results are displayed in Figure 9. The total length of the topological map is 0.67 km, and there are 147 vertices in total. Furthermore, we ignore people, moving cars, and bicyclists in the perception data for the completeness of the map. Figure 10 displays the routing results using our maps with random starting and ending points. The average time cost in A* path searching at 470 iterations is 0.55 ms, which is more efficient than the algorithm in metric maps, such as A*, RRT, and BIT* [17]. The above experiments demonstrate the capability of our method in complex real-world scenarios.

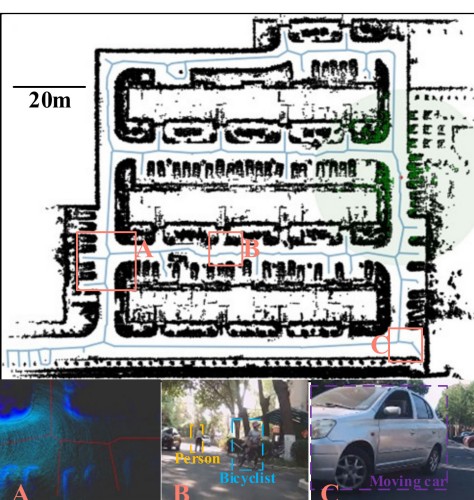

**Figure 9.** A topological map and a metric map in the real world. the A, B and C are labeled areas. A is a superimposed point cloud at a road intersection; B and C are images with ignored objects from perception data.

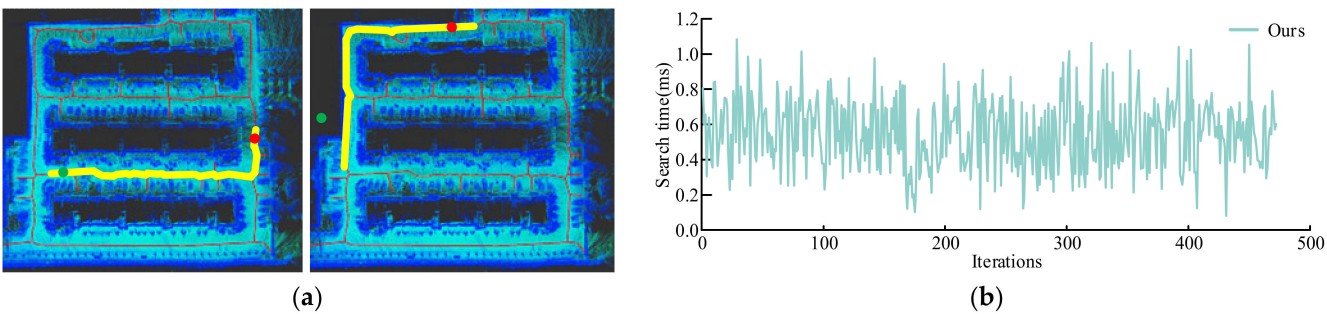

(**a**)                         (**b**)

**Figure 10.** Path-searching results with A*. (**a**) The yellow paths that start from the red circle to the green circle, and the goal in the right diagram is out of the road. (**b**) The path-searching time cost in random tests.

## 5. Conclusions

In this paper, we propose a compact and efficient topological mapping method based on the obstacle points from LiDAR. First, the sweep line algorithm is used to generate a Voronoi diagram from the obstacle information in the environment. Then, to obtain a more compact map, we use a pruning strategy to delete redundant vertices and edges in the graph. Furthermore, to keep it efficient when working in large-scale scenarios, we propose a new incremental updating method for Voronoi diagrams based on the property of the largest empty circle to accurately update the changed regions and their boundaries. Our method is evaluated against state-of-the-art methods in simulation environments, and the results show that our method generates a much more compact topological map while consuming the lowest average computation. The real-world tests also demonstrate the competence of our method. In the future, we will further extend the method on remote sensing images from UAVs.

**Author Contributions:** Conceptualization, Y.Q. and R.W.; methodology, Y.Q. and B.H.; software, Y.Q.; validation, Y.Q., B.H. and F.L. formal analysis, Y.Q., B.H. and R.W.; writing—original draft preparation, Y.Q. and R.W.; writing—review and editing, Y.Q., Y.X. and R.W.; project administration, Y.X.; funding acquisition, Y.X. All authors have read and agreed to the published version of the manuscript.

**Funding:** This research was funded by the National Key Research and Development Program of China, grant number 2016YFB0100903.

**Institutional Review Board Statement:** Not applicable.

**Informed Consent Statement:** Not applicable.

**Data Availability Statement:** All data can be requested from the corresponding author.

**Acknowledgments:** The authors would like to thank the handling editor and the anonymous reviewers for their valuable comments and suggestions.

**Conflicts of Interest:** The authors declare no conflict of interest.

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
