# Peer review of "Compact and Efficient Topological Mapping for Large-Scale Environment with Pruned Voronoi Diagram"

_drones, doi:10.3390/drones6070183_

Round 1

Reviewer 1 Report

 Authors present a compact and efficient topological mapping method for large-scale environment. The paper is interesting, and it will be helpful for further research. The methodology description is presented systematically and results are well compared. Some comments are:

- In the introduction part, authors mentioned a efficiency of map update in their contributions. Although it is explained in the following sentence, it is necessary to specify what kind of efficiency it is for clarity. 

- Line 71: How is the random increment algorithm one of the most efficient?

- Line 128: How does post processing affect real-time performance? 

- It's desirable to have consistency in the expression when citing references. 

- Figure 8: It would be better if a figure that could compare the results more visibly is added. 

- The language should be accurately revised. There are incomplete sentences. 

Author Response

Dear Professor,

    Thank you very much for your time involved in reviewing the manuscript and your very encouraging comments on the merits. Please see the attachment for responses.

   Sincerely,

   The Authors

Reviewer 2 Report

General comments

The paper entitled “Compact and Efficient Topological Mapping for Large-scale Environment with Pruned Voronoi Diagram” treats about a topic of the highest interest in the robot autonomous navigation planning scope. The manuscript introduces modification to the classical Voronoi diagram with the aim of producing a compact topological map. The task is accomplished by means of a three steps procedure. The three step procedure description is rather difficult to follow. This reviewer guesses that it can be improved by using some flow- diagrams. Authors are warmly encouraged to add those diagrams. The declared novelty of the manuscript is the introduction of the reduced approximated generalized Voronoi graph, whose effectivity is proved by comparing it with the outcomes of other state of art similar methodology. There are no reference for SLAM.

The used language was pertinent, technically sound and seamless. The paper is linear and well-structured. Unfortunately, the heavy technicalities adopted don’t allow not expert users to follow easily the treated argument. The typographical outline is effective and satisfactory making the paper highly readable. Key Words are pertinent to the paper content and appropriate. The highlights are not included within the paper, therefore no judgement has been provided about them. The References section is rich both in quantity and quality. Graphic representations are all of good quality and self-explaining. Finally, the treated topic falls under the journal scope without any doubt. This reviewer considers the paper worthy of publication after minor changes.

Specific comment

Line 40: maybe, the sentence should be re-written it appears to be unclear.

Line 42: maybe, Authors intended “variants” and not “variances”.

Line 99: maybe, Authors intended “some posed drifts”?

Line 137: the term “Coccupied” appears to be incorrect.

Author Response

(The authors gave the same response as above.)

Reviewer 3 Report

The article is well-written and easy to follow. The authors present new findings while their experimental protocol both in a simulation environment and the real-world case proves the system's significance. 

Author Response

(The authors gave the same response as above.)
